# A kinesin Klp10A mediates cell cycle-dependent shuttling of Piwi between nucleus and nuage

Zsolt G. Venkei[1], Charlotte P. Choi[2], Suhua Feng[3,4], Cuie Chen[1], Steven E. Jacobsen[3,4,5], John K. Kim[2], Yukiko M. Yamashita[1,6,7] *

**1** Life Sciences Institute, University of Michigan, Ann Arbor, Michigan, United States of America,
**2** Department of Biology, Johns Hopkins University, Baltimore, Maryland, United States of America,
**3** Department of Molecular, Cell and Developmental Biology, University of California, Los Angeles, California, United States of America, **4** Eli and Edyth Broad Center of Regenerative Medicine and Stem Cell Research, University of California, Los Angeles, California, United States of America, **5** Howard Hughes Medical Institute, University of California, Los Angeles, California, United States of America, **6** Department of Cell and Developmental Biology, University of Michigan, Ann Arbor, Michigan, United States of America, **7** Howard Hughes Medical Institute, University of Michigan Ann Arbor, Michigan, United States of America

* yukikomy@umich.edu

**Data Availability Statement:** All data are available within the manuscript except for the sequencing data, which is available at GEO: accession number: GSE122596.

## Abstract

The piRNA pathway protects germline genomes from selfish genetic elements (e.g. transposons) through their transcript cleavage in the cytoplasm and/or their transcriptional silencing in the nucleus. Here, we describe a mechanism by which the nuclear and cytoplasmic arms of the piRNA pathway are linked. We find that during mitosis of *Drosophila* spermatogonia, nuclear Piwi interacts with nuage, the compartment that mediates the cytoplasmic arm of the piRNA pathway. At the end of mitosis, Piwi leaves nuage to return to the nucleus. Dissociation of Piwi from nuage occurs at the depolymerizing microtubules of the central spindle, mediated by a microtubule-depolymerizing kinesin, Klp10A. Depletion of *klp10A* delays the return of Piwi to the nucleus and affects piRNA production, suggesting the role of nuclear-cytoplasmic communication in piRNA biogenesis. We propose that cell cycle-dependent communication between the nuclear and cytoplasmic arms of the piRNA pathway may play a previously unappreciated role in piRNA regulation.

## Author summary

The piRNA pathway that defends germline from selfish elements operates in two subpathways, one mediated by Piwi in *Drosophila* to silence transcription of targets in the nucleus and the other mediated by Aub and Ago3 to cleave transcripts of targets in the cytoplasm. How these two subpathways might coordinate with each other, particularly at the cell biological level, remains elusive. This study shows that Piwi interacts with Aub/Ago3 specifically in mitosis in nuage, the organelle that serves as the platform for piRNA cytoplasmic subpathway. Piwi returns to the nucleus at the end of mitosis, and our study suggests that dissociation of Piwi from nuage is facilitated by microtubule depolymerization by a

**Funding:** This work was supported by Howard Hughes Medical Institute to YMY and SEJ (http://www.hhmi.org), and National Institute of General Medical Sciences (NIH R01GM118875) to JKK (https://www.nigms.nih.gov). The funders had no role in study design, data collection and analysis, decision to publish, or preparation of the manuscript.

**Competing interests:** No competing interests.

kinesin Klp10A at the central spindle. We propose that cell-cycle-dependent interaction of two piRNA subpathways may play an important role in piRNA production.

## Introduction

Piwi-associated RNAs (piRNAs), a class of endogenous small RNAs found across many organisms, associate with Piwi proteins of the Argonaute family to silence active transposable elements (TEs)[1–5]. In *Drosophila melanogaster*, Piwi, Aubergine (Aub), and Argonaute 3 (Ago3) comprise the three Piwi proteins of the piRNA pathway [1, 6–8]. The piRNA pathway is initiated in the nucleus, where primary piRNA precursors are transcribed from genomic piRNA clusters, translocated to the cytoplasm, and processed into mature piRNAs [1, 9]. In the cytoplasm of the germline, primary piRNAs are amplified in a "ping-pong" cycle that simultaneously cleaves TEs and produces secondary piRNAs [7, 10, 11]. Specifically, primary piRNAs antisense to TEs are loaded into and direct Aub to cleave TE transcripts, which produces secondary piRNAs that are subsequently loaded into Ago3. The secondary piRNAs generated by the ping-pong cycle are then loaded into Piwi and translocated into the nucleus as a piRNA-induced silencing complex (piRISC) [6, 12]. Within the nucleus, Piwi leads to the transcriptional repression of TEs through the deposition of heterochromatic histone marks [13–15]. Thus, the cytoplasmic arm of the piRNA pathway intimately interacts with the nuclear arm to affect silencing at the post-transcriptional and transcriptional levels, respectively.

As evidence of the tight coupling between the two arms of the piRNA pathway, ping-pong amplification takes place in nuage–the electron-dense, phase-separated granules that are anchored directly to the cytoplasmic face of the nuclear pore [16–18]. The interface between nuage and the nucleus has been shown to be critical for piRNA biogenesis and transposon target recognition and repression. Although nuage has been characterized as a static and long-term platform for piRNA biogenesis, individual nuage components, such as Aub and Ago3, are dynamically shuttled in and out of nuage [1, 6, 19–21]. Unlike Aub and Ago3, Piwi shows predominantly nuclear localization [22] and has not been well-characterized within the context of nuage. Although previous studies have identified multiple piRNA pathway components that are required for proper localization of piRNA amplification machinery to nuage, including Spindle-E, Qin, Krimper, and Vasa, the underlying mechanism by which ping-pong-amplified piRNAs translocate into the nucleus remains poorly understood [6, 16, 21, 23–25].

Here we show that nuclear and cytoplasmic components of the piRNA pathway interact specifically during mitosis of *Drosophila* spermatogonia (SGs) to likely facilitate communication between the nuclear and cytoplasmic arms of the pathway. We found that Piwi localizes to nuage specifically during mitosis, and returns to the nucleus upon mitotic exit. Through immunoprecipitation and mass spectrometry, we found that Klp10A, a microtubule (MT)-depolymerizing kinesin motor protein [26], physically interacts with components of the piRNA pathway. In the absence of Klp10A, Piwi remains in nuage for a prolonged period after mitotic exit, thus delaying its return to the nucleus. Cytological observations suggest that dissociation of Piwi from nuage at the end of mitosis occurs at the central spindle while it is being depolymerized in a *klp10A*-dependent manner. *klp10A*-depletion leads to increased piRNA production, supporting the physiological role of Klp10A-mediated regulation of Piwi localization. We propose that the interaction of Piwi with nuage during mitosis may represent a previously unappreciated mechanism to regulate piRNA biogenesis.

## Results

### Klp10A physically interacts with piRNA pathway components in germline stem cells and spermatogonia of *Drosophila* testis

*Drosophila* spermatogenesis is supported by asymmetrically dividing germline stem cells (GSCs), which produce a self-renewing GSC and a gonialblast (GB) that initiates the differentiation program. GBs further undergo four rounds of mitotic divisions as spermatogonia (SGs), which then enter meiotic program as spermatocytes (SCs) (Fig 1A). Asymmetric divisions of *Drosophila* male GSCs, as well as several other systems, are mediated by stereotypical positioning of the centrosomes [27–36]. Therefore, stem cell-specific centrosomal components are of significant interest in understanding the mechanisms that regulate asymmetric stem cell divisions.

In a previous study, we showed that Klp10A protein is enriched on the centrosomes specifically in GSCs, but not in differentiating SGs [37] (Fig 1A). In an attempt to isolate proteins that specifically localize to GSC centrosomes, we affinity-purified Klp10A from a GSC-enriched extract, using either specific antibody or anti-GFP antibody combined with expression of Klp10A-GFP (see Methods). To enrich for GSCs, the self-renewal factor Unpaired (Upd) was expressed (*nos-gal4>UAS-upd*). Upd is a ligand expressed by hub cells to activate the JAK-STAT pathway, which determines GSC self-renewal. It was shown that ectopic expression of Upd in germ cells (*nos-gal4>UAS-upd*) is sufficient to cause GSC-like tumors in the testis [38, 39]. Anti-Klp10A and anti-GFP immunoprecipitates were analyzed by mass-spectrometry (see Methods). Many MT-associated proteins were enriched in Klp10A pull-down samples (S1–S3 Tables), validating the specificity of the Klp10A pull down experiments.

To our surprise, we found that components of the piRNA pathway, such as Vasa, Aub and Piwi, were enriched in the Klp10A pull-downs (S1–S3 Tables, Table 1). The fact that many piRNA pathway components were consistently identified in Klp10A-pulldowns led us to speculate this interaction might be of significance. Indeed, immunoprecipitation using anti-Klp10A antibody and GSC-enriched extracts confirmed that Klp10A physically interacts with Vasa (Fig 1B). Similarly, anti-GFP antibody pull down of GFP-Aub confirmed the Aub-Klp10A interaction (Fig 1C). Although it remains elusive whether Klp10A directly binds to these proteins or perhaps through an intermediary such as RNA, the fact that Klp10A co-immunoprecipitates multiple piRNA pathway components indicates that their interaction may be of biological relevance, regardless of whether the interaction is direct or not. Similar interactions between Klp10A and piRNA pathway components (Vasa, Piwi, Aub, Ago3) were observed when extracts from SG tumor was used (*nos>dpp*, [40–42]) (Fig 1D and 1E, S4 and S5 Tables, Table 1), suggesting that the Klp10A interaction with piRNA pathway components is not unique to GSCs. Cytological data confirmed that Klp10A interaction with piRNA pathway components is not limited to GSC but also occurs in SGs (see below). Accordingly, the study diverged from our initial intention to isolate GSC centrosome-specific components. However, we investigated the unexpected role of *klp10A* in the regulation of the piRNA pathway, as the study revealed an unappreciated mode of regulation of piRNA biogenesis.

### Depletion of *klp10A* results in alteration of piRNA biogenesis

To explore whether Klp10A interaction with piRNA pathway components was functionally significant, we examined whether *klp10A* was required for piRNA biogenesis [43]. To characterize the role of *klp10A* in the piRNA pathway, we first performed deep sequencing of small RNAs from wild type testes or testes with germline-specific *klp10A*-knockdown (*nos-gal4>UAS-klp10A*^TRiP.HMS00920^, validated in our previous study [37], and hereafter referred to

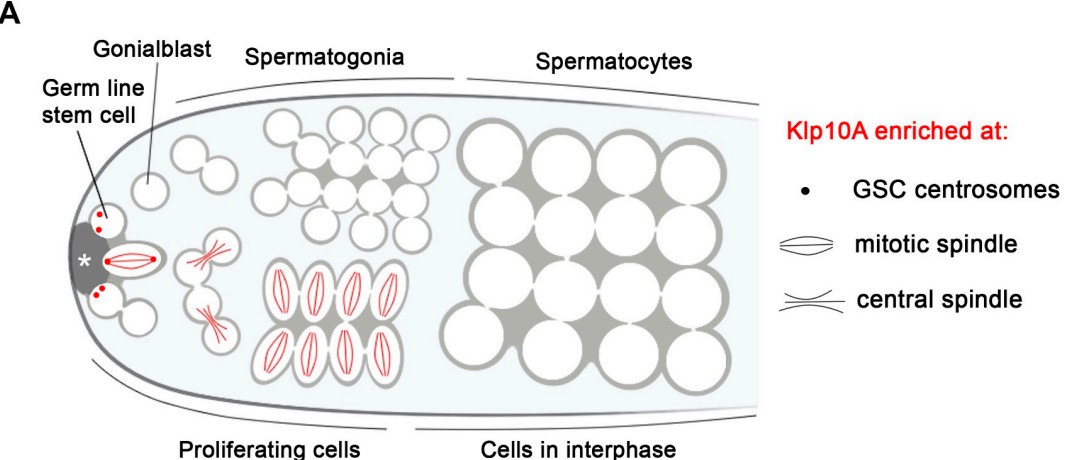

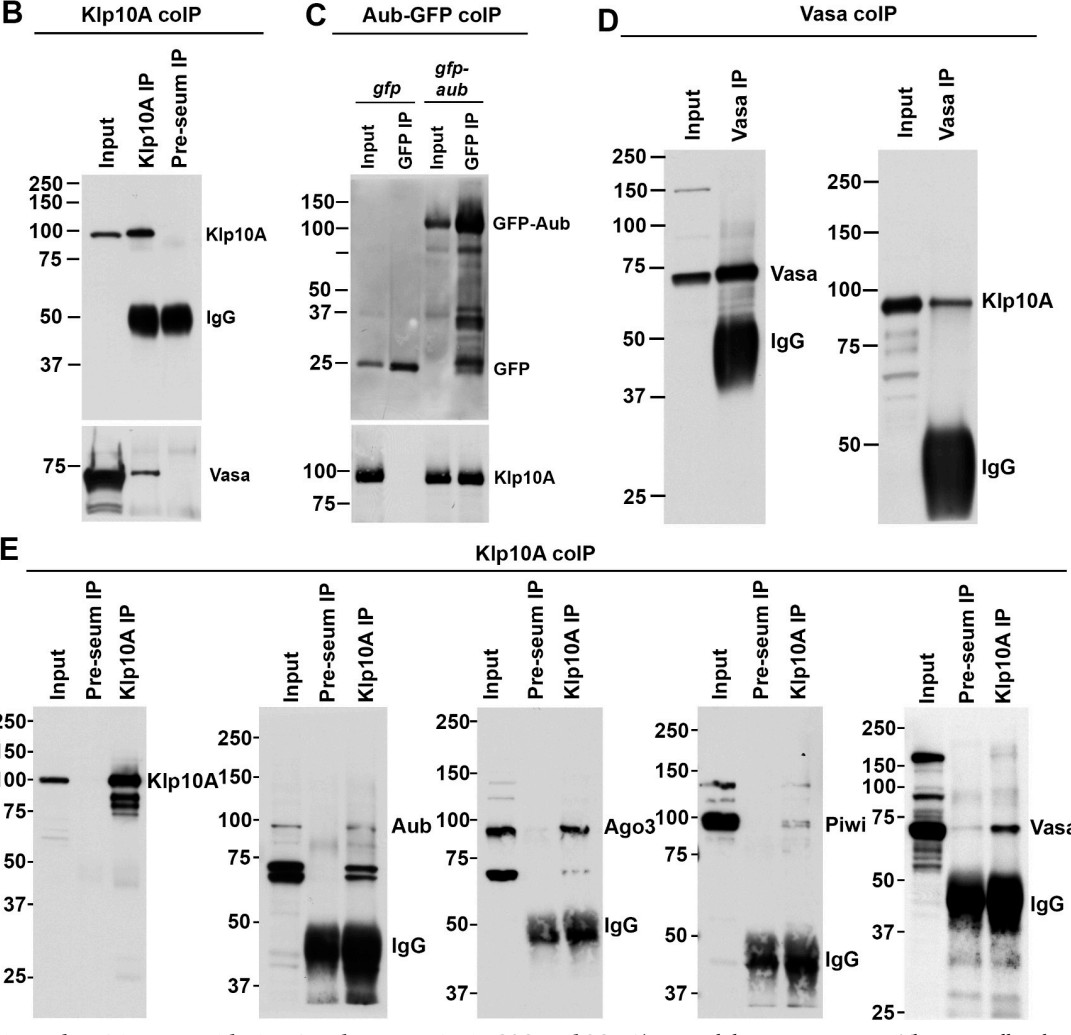

**Fig 1. Klp10A interacts with piRNA pathway proteins in GSCs and SGs.** A) *Drosophila* spermatogenesis. The stem cell niche is formed by non-dividing somatic cells (hub, marked by asterisk). Germline stem cells (GSCs) are physically attached to the hub, and divide asymmetrically. Gonialblasts (GBs), the differentiating daughters of GSCs, undergo four rounds of mitotic divisions with incomplete cytokinesis. Resultant 16-cell spermatogonia (SGs) then enter meiotic prophase as spermatocytes (SCs). Klp10A is specifically enriched at the centrosomes of GSCs (indicated by red dots), and at the central spindles of GSCs/ GBs/SGs (red lines) [37]. B) Vasa co-immunoprecipitates with Klp10A in the extract from Upd-expressing testes (*nos-gal4>UAS-upd*). L: lysate input, IP: immunoprecipitant. C) Klp10A co-immunoprecipitates with GFP-Aub in the extract from

Upd-expressing testes (*nos-gal4>UAS-upd UAS-gfp*, vs. *nos-gal4>UAS-upd UAS-gfp-aub*). D) Klp10A co-immunoprecipites with Vasa in the extract from the Dpp-expressing testes (*nos-gal4>UAS-dpp*). E) piRNA pathway proteins Aub, Ago3, Piwi and Vasa co-immunoprecipitate with Klp10A in the extract from Dpp-expressing testes (*nos-gal4>UAS-dpp*).

as *klp10A^RNAi*) (S1A Fig). We defined piRNAs as reads of 23–29 nucleotides (nt) in length that did not map to microRNAs or ribosomal RNAs. The majority of reads that are 23–29 nucleotides in length start with a Uridine at the 5' most position, which has shown to be a unique, conserved feature of mature piRNAs [44], whereas reads that are 20–22 nucleotides in length, defined as siRNAs in our study, do not show such enrichment (S2 Fig), validating our analysis.

To assess the global changes in piRNA expression upon *klp10A* loss, we profiled piRNA reads across the *Drosophila* transcriptome. *klp10A^RNAi* caused specific upregulation of piRNAs mapping to repetitive elements and piRNA clusters, with minimal differences in piRNAs mapping to other genomic classes, or in other types of small RNAs such as microRNAs (Fig 2A, S3 Fig). These data suggest that *klp10A* may play a specific role in the piRNA pathway. When we analyzed the abundance of piRNAs mapping to TEs upon loss of *klp10A*, we observed significant upregulation of piRNA reads both antisense and sense to TEs (Fig 2B–2E) without any noticeable changes in their distribution (Fig 2F, S4 Fig). The total amount of piRNA per TE was upregulated up to ~4 times (Fig 2B), with upregulation of individual piRNA being more prominent (Fig 2C, S4 Fig).

The above data showed that both primary and secondary piRNAs were upregulated, suggesting that the ping-pong cycle itself remains intact in *klp10A^RNAi*. To test this, we calculated the number of piRNAs with a ping-pong signature (i.e. antisense and sense read pairs that have 10-nucleotide complementary base pairing from their 5' ends) in both wildtype and *klp10A^RNAi* libraries (S5A Fig). We observed a significant bias for piRNA read pairs with 10-nucleotide complementarity in both wildtype and *klp10A^RNAi*, suggesting that the ping-pong pathway indeed remains active upon loss of *klp10A*. Moreover, we found no significant changes in ping-pong ratios in *klp10A^RNAi* compared to wildtype (S5B Fig), further confirming an intact ping-pong cycle.

To address if piRNA upregulation in *klp10A^RNAi* leads to changes in TE expression, we performed mRNA-sequencing in wildtype and *klp10A^RNAi* germ cells (S6 Fig). We found that most TEs did not show any significant changes in expression levels except for a small subset of TEs, which showed moderate downregulation upon loss of *klp10A* (Fig 2D). There were no clear common characteristics among these TEs that may explain as to why these TEs exhibits downregulation. Taken together, our results show that *klp10A* is involved in piRNA production and that Klp10A interaction with piRNA pathway components may have functional significance (see Discussion), prompting us to further examine the underlying mechanisms. It should be noted that *klp10A^RNAi* does not lead to noticeable changes in overall cellular composition of the testes, the process of differentiation [37], or cell cycle distribution (see below). Therefore, it is unlikely that skewed composition of cell types/cell cycle is the cause of observed changes in piRNA levels in *klp10A^RNAi*.

## Klp10A colocalizes with piRNA pathway components at the central spindle during telophase in GSCs and SGs

To begin to explore how Klp10A might contribute to piRNA biogenesis, we examined the localization of Klp10A protein and piRNA pathway components throughout the cell cycle of GSCs and SGs. Vasa and Aub showed well-established perinuclear localization in nuage throughout interphase (Fig 3A, 3B, 3E and 3F) [8, 45, 46]. During this period, Klp10A showed centrosome localization in GSCs (Fig 3A, 3E and 3I), or uniform cytoplasmic localization in

**Table 1. Enrichment of piRNA pathway proteins in GFP-Klp10A and Klp10A pull downs from GSC and SG tumor testes.** PSM: number of peptide spectrum matches. Ranking indicates PSM number-based ranking of the protein/all identified proteins of the pull down. piRNA pathway proteins that ranked among top 20% are indicated by bold.

| | Control pull down | | Klp10A pull down | | | |
| --- | --- | --- | --- | --- | --- | --- |
| | Pulled down with GFP from Upd testis | | Pulled down with GFP-Klp10A from Upd testis | | | |
| | PSM | Ranking | PSM | Ranking | | |
| **Aubergine** | 1 | 31/33 | **9** | **63/742** | | |
| **Vasa** | 2 | 13/33 | **9** | **64/742** | | |
| Piwi | 0 | NA | 4 | 185/742 | | |
| Spindle-E | 0 | NA | 2 | 346/742 | | |
| Tudor | 0 | NA | 2 | 351/742 | | |
| Shutdown | 0 | NA | 2 | 387/742 | | |
| Armitage | 0 | NA | 1 | 603/742 | | |
| Vrento | 0 | NA | 1 | 704/742 | | |
| Krimper | 0 | NA | 1 | 718/742 | | |
| | Pulled down with pre-immune IgG from Upd testis | | Pulled down with Klp10A from Upd testis | | | |
| | | | Experiment 1 | | Experiment 2 | |
| | PSM | Ranking | PSM | Ranking | PSM | Ranking |
| **Aubergine** | 0 | NA | **53** | **89/1318** | **92** | **67/1456** |
| **Piwi** | 0 | NA | **27** | **189/1318** | **66** | **97/1456** |
| Argonaute 3 | 0 | NA | 0 | NA | 22 | 323/1456 |
| Spindle-E | 0 | NA | 11 | 448/1318 | 21 | 337/1456 |
| Vasa | 0 | NA | 4 | 833/1318 | 20 | 358/1456 |
| Hsp90 | 0 | NA | 17 | 318/1318 | 6 | 834/1456 |
| Armitage | 0 | NA | 6 | 674/1318 | 13 | 551/1456 |
| Qin | 0 | NA | 11 | 443/1318 | 8 | 705/1456 |
| Tudor | 0 | NA | 5 | 731/1318 | 10 | 641/1456 |
| Tapas | 0 | NA | 9 | 500/1318 | 8 | 711/1456 |
| UAP56 | 2 | 38/38 | 3 | 973/1318 | 10 | 631/1456 |
| Papi | 0 | NA | 0 | NA | 2 | 1444/1456 |
| | | | Pulled down with Klp10A from Dpp testis | | | |
| | | | Experiment 1 | | Experiment 2 | |
| | | | PSM | Ranking | PSM | Ranking |
| **Aubergine** | | | **127** | **22/1019** | **129** | **27/924** |
| **Qin** | | | **36** | **148/1019** | 0 | NA |
| **Piwi** | | | **32** | **164/1019** | **25** | **166/924** |
| **Vasa** | | | 16 | 300/1019 | 28 | 152/924 |
| **Tapas** | | | **27** | **192/1019** | 0 | NA |
| Tudor | | | 24 | 220/1019 | 9 | 446/924 |
| Argonaute 3 | | | 11 | 404/1019 | 15 | 266/924 |
| Spindle-E | | | 15 | 314/1019 | 4 | 635/924 |
| Armitage | | | 9 | 456/1019 | 0 | NA |
| Hsp90 | | | 8 | 471/1019 | 5 | 591/924 |
| UAP56 | | | 7 | 544/1019 | 5 | 492/924 |
| Maelstrom | | | 2 | 1003/1019 | 0 | NA |

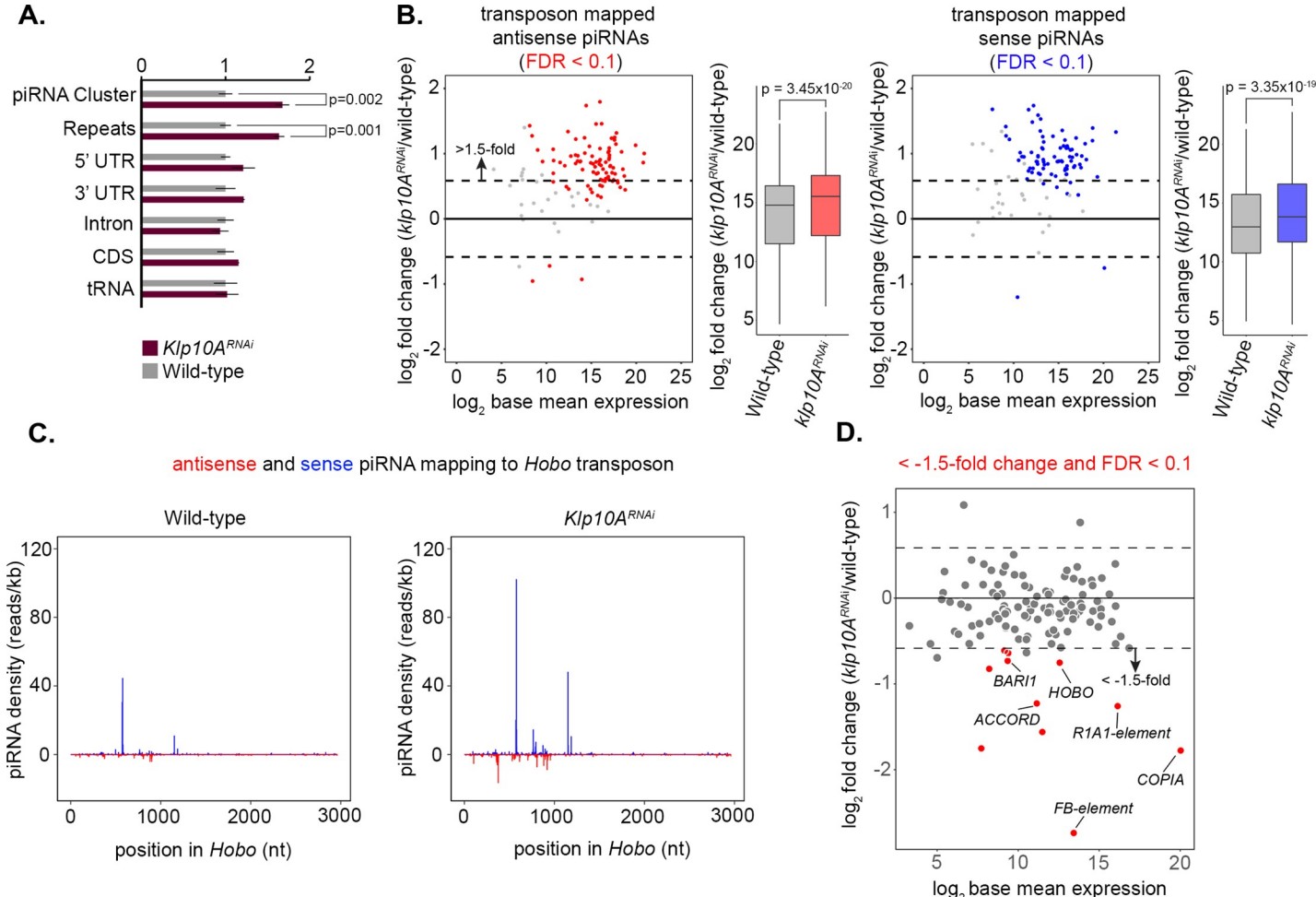

**Fig 2. piRNA production is upregulated in *klp10A^RNAi* testis.** A) piRNA reads normalized to total library reads at each genomic feature. piRNAs are significantly upregulated in repeat and piRNA cluster regions. Significance was calculated using a Student's t-test. B) Scatter and bar plots comparing transposon mapped sense and antisense piRNA abundance in *klp10A^RNAi* germ cells vs. wild-type. piRNA abundance was calculated by normalizing piRNA read counts mapping to each transposon to total miRNA hairpin reads. A change of greater than 1.5-fold and FDR of <0.1 were used as cutoffs for differential analysis. Significance for piRNA fold change expression was calculated using Wilcoxon signed-rank test. C) Density of sequenced piRNAs (blue: sense; red: antisense) across *HOBO*. Each piRNA read was normalized to miRNA hairpin abundance. D) Scatter plot comparing transposon abundance in *klp10A^RNAi* versus wild-type (red: FDR<0.1).

SGs (Fig 3B, 3F and 3J) as we reported previously [37]. We did not observe clear co-localization between centrosomal Klp10A and nuage, suggesting that the potential role of Klp10A in piRNA pathway is not related to its GSC-specific centrosomal localization.

When cells enter mitosis, nuage became somewhat larger in size, which we refer to as 'mitotic nuage' hereafter (Fig 3C, 3G and 3K). Mitotic nuage appears to be the same population described previously as 'Vasa granule near the mitotic chromosomes' by Pek and Kai [47]. At this point, Klp10A was observed at the spindle pole (and weakly on the spindle) as previously reported [37], showing little colocalization with Vasa/Aub (Fig 3C, 3G, 3K and 3M–3O).

Although Klp10A did not noticeably colocalize with any of nuage components until anaphase, colocalization became clear during telophase: nuage was associated with Klp10A near the center of the dividing cell (Fig 3D, 3H and 3L–3O), which we confirmed to be bundles of central spindle MTs (S7 Fig). Importantly, Klp10A localization to the central spindle was observed both in GSCs and SGs [37], and colocalization of Klp10A and piRNA pathway components was commonly observed both in GSCs and SGs.

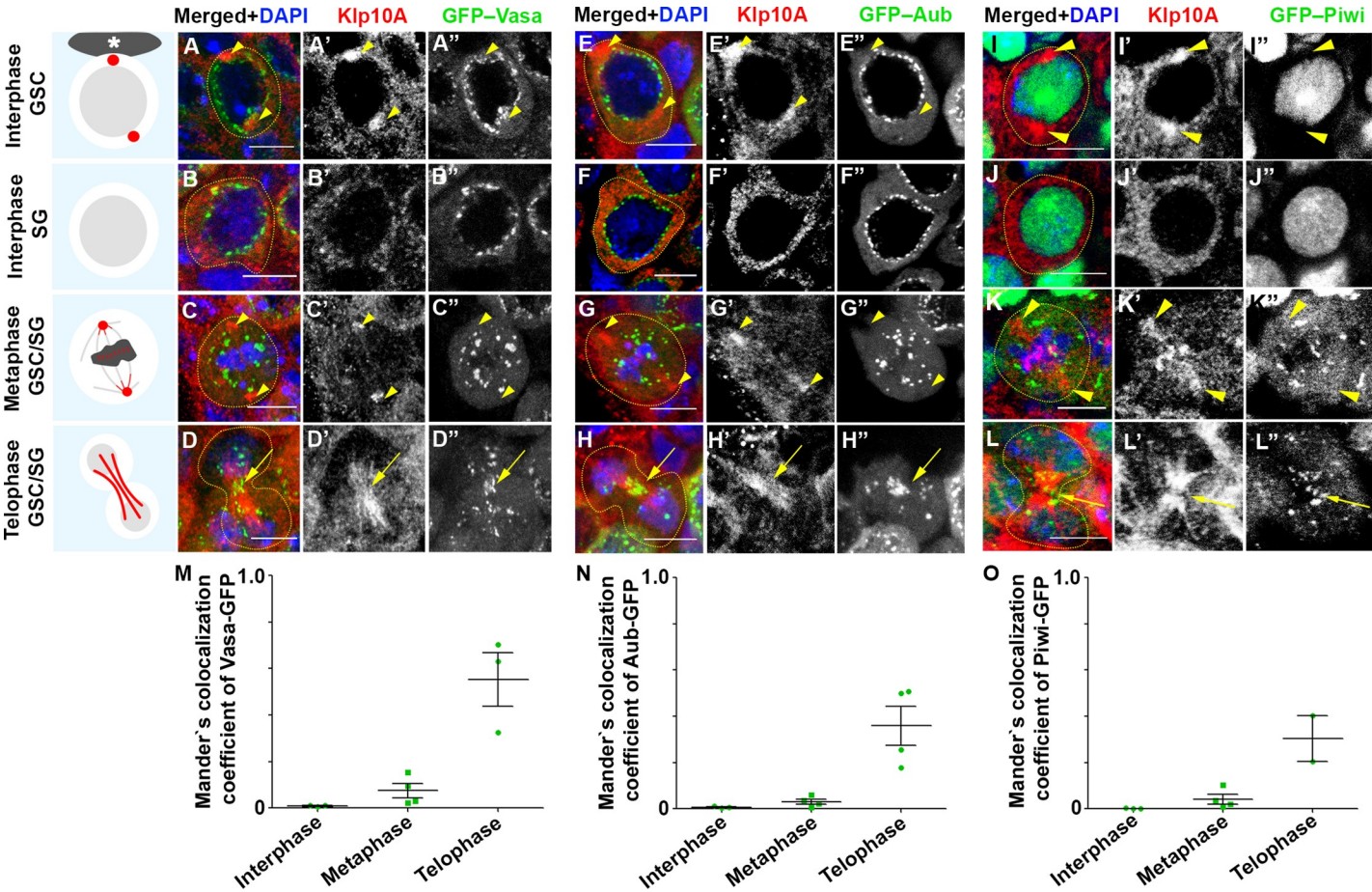

**Fig 3. Klp10A colocalizes with Vasa, Aub and Piwi at the central spindle in GSCs/SGs during mitotic exit.** A-D") Localization of Klp10A (red) and GFP-Vasa (green) in GSCs/SGs throughout the cell cycle. DAPI (blue). E-H") Localization of Klp10A (red) and GFP-Aub (green) in GSCs/SGs throughout the cell cycle, I-L") Localization of Klp10A (red) and Piwi-GFP (green) in GSCs/SGs throughout the cell cycle. Centrosomal localization of Klp10A is indicated by arrowheads, central spindle localization with arrows. Bars: 5 µm. M-O) Quantification colocalization of GFP-Vasa (M), GFP-Aub (L) and GFP-Piwi (O) with Klp10A in GSCs/SGs during cell cycle. Mender's colocalization coefficient is calculated based on how much of indicated proteins colocalize with Klp10A. Data points represent single cells, each from different testes. Error bars represent SD.

In contrast to Aub and Ago3, which reside in nuage together with Vasa to function in the cytoplasmic arm of the piRNA pathway, Piwi functions in the nucleus to repress TEs at the transcriptional level [15, 16, 43, 48–50]. We confirmed that GFP-Piwi localizes to the nucleus during interphase of GSCs and SGs as described previously [22, 51](Fig 3I and 3J). Interestingly, we found that once cells enter mitosis, GFP-Piwi localized to mitotic nuage together with the components of the cytoplasmic arm of the piRNA pathway, such as Aub and Vasa (Fig 3K, see Fig 4 for the confirmation of Piwi's localization to mitotic nuage). During telophase, GFP-Piwi still colocalized with nuage components at the central spindle (Fig 3L and 3O), after which it returned to the nucleus. Taken together, these data show that piRNA pathway components likely interact with Klp10A at the central spindle during telophase.

## Klp10A is required for relocation of Piwi from mitotic nuage to the nucleus at the end of mitosis

Based on the above results that suggest Klp10A interaction with nuage components at the central spindle, we explored how this interaction may impact the piRNA pathway. To this end, we first examined the localization of Piwi and Vasa in control vs. *klp10A^{RNAi}* GSCs/SGs.

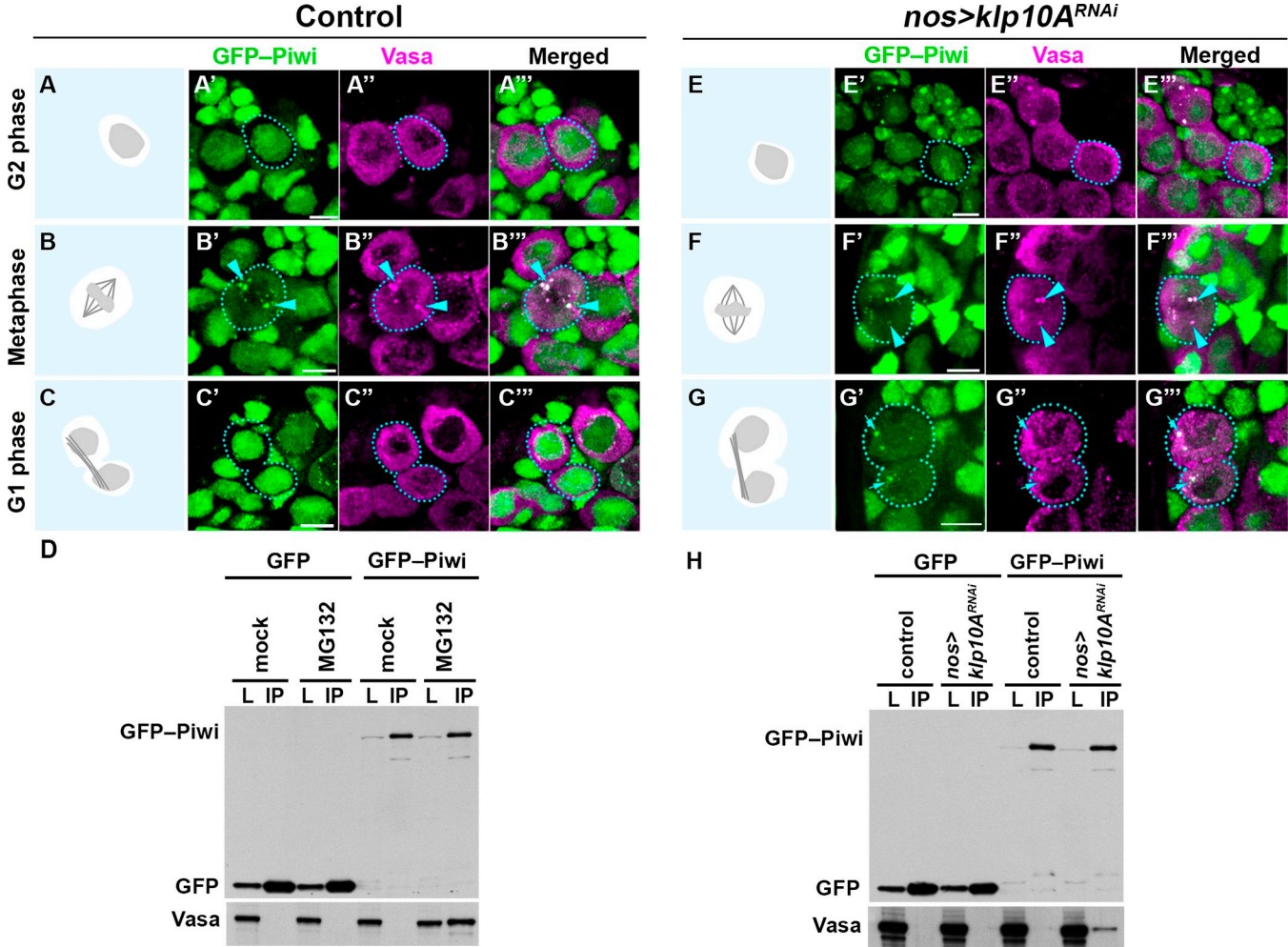

**Fig 4. Piwi interacts with cytoplasmic piRNA pathway components at the nuage in mitosis, and returns to the nucleus at the end of mitosis in a Klp10A-dependent manner.** (A-C) GFP-Piwi (green) and Vasa (magenta) localization throughout the GSC cell cycle in wild type testes. The localization patterns were the same in GSCs and SGs. GSCs are encircled by dotted lines. Cartoons show DAPI and MT morphology (shown in S8 Fig) to assess cell cycle stages. GFP-Piwi and Vasa colocalization at the mitotic nuage is indicated by arrowheads. Bars: 5 μm. D) co-immunoprecipitation of Vasa with GFP-Piwi from *nos>upd* testes after 4h 30min *ex vivo* MG132 treatment. *nos>upd*, *gfp* testes were used as control. L: lysate input, IP: immunoprecipitant. (E-G) GFP-Piwi (green) and Vasa (red) localization throughout the GSC cell cycle in *klp10A^RNAi* testes. Cytoplasmic GFP-Piwi/Vasa granule in post-mitotic interphase cells are highlighted by arrows in panel G. H) co-immunoprecipitation of Vasa with GFP-Piwi from *nos>upd* testes after *klp10A^RNAi*. *nos>upd*, *gfp* and *nos>upd*, *gfp*, *klp10A^RNAi* testes were used as control.

We confirmed the above observation that Piwi localizes to mitotic nuage by examining the colocalization of GFP-Piwi with Vasa in wild type cells. Although their localization was distinct during interphase (i.e. Vasa in nuage, Piwi in the nucleus) (Fig 3, Fig 4A and 4C, S8 Fig), Vasa and Piwi colocalized during mitosis (Fig 4B, S8 Fig, S9 Fig), and Piwi returned to the nucleus at the end of mitosis (Fig 4C, S10 Fig). We further detected physical interaction and colocalization between Vasa and Piwi specifically when cells were blocked in metaphase with MG132 (a proteasome inhibitor that arrests cells in mitosis [52]), or with colcemid (S9 Fig) [31]. These results suggest that Piwi gains access to nuage specifically during mitosis.

Next, we investigated whether *klp10A^RNAi* testes show any defects in the behavior of nuage or Piwi during the cell cycle. Nuage morphology and composition appeared unperturbed in most interphase cells (Fig 4E), and in all mitotic cells until anaphase in both GSCs and SGs of

*klp10A^RNAi* testes (Fig 4F, S10A Fig). However, the difference between the control and *klp10A^RNAi* germ cells became clear upon exiting mitosis ('G1' in Fig 4C and 4G). In control germ cells, Piwi returned to the nucleus at the end of telophase and became exclusively nuclear in the subsequent interphase as described above (Fig 4C, S10 Fig, S11A Fig). In contrast, in *klp10A^RNAi* germ cells, Piwi persisted in cytoplasmic nuage even after the completion of mitosis (Fig 4G, S10 Fig, S11B and S11C Fig). Piwi-Vasa interaction was detectable by co-immunoprecipitation in *klp10A^RNAi* without being arrested in mitosis (Fig 4H), consistent with persistent colocalization of Piwi with Vasa at nuage in early G1/S cells (Fig 4G, S11B and S11C Fig). Likely reflecting the delayed/incomplete translocation of Piwi from mitotic nuage to the nucleus, we also observed that the amount of nuclear Piwi was slightly but significantly reduced in GSCs/SGs in *klp10A^RNAi* germ cells (S12 Fig). Delayed dissociation of Piwi from mitotic nuage at the end of mitosis was confirmed by live observation of GFP-Piwi and mCherry-Vasa expressed in mitotic SGs (S1 and S2 Videos). Taken together, these results show that 1) Piwi interacts with nuage specifically in mitosis, and 2) *klp10A* is required for releasing Piwi from mitotic nuage to allow its return to the nucleus at the end of mitosis.

## Piwi dissociates from nuage at the central spindle microtubules

How does Piwi dissociate from mitotic nuage at the end of mitosis, and how does *klp10A* promote this process? A closer examination of live imaging using GFP-Piwi and mCherry-Vasa (S1 Video) revealed that a Piwi-positive compartment ('Piwi granule' hereafter) budded off from mitotic nuage and then dispersed near the center of telophase cells at the central spindle (see below for the confirmation of their localization at the central spindle) (Fig 5A, S1 Video). Concomitantly, nuclear Piwi level gradually increased (S1 Video), suggesting that Piwi released from mitotic nuage returned to the nucleus. Morphology of nuage marked by Vasa remained unchanged during this period (S1 Video). In contrast to control SGs, we barely observed the budding off of the Piwi granule in *klp10A^RNAi* SGs (Fig 5B, S2 Video). As a result, Piwi remained in nuage for a prolonged time period even after the completion of mitosis. These results show that *klp10A* is required for releasing Piwi from mitotic nuage such that Piwi can return to the nucleus. It should be noted that *klp10A^RNAi* germ cells did not exhibit noticeable change in cell cycle progression (see below). Therefore, it is unlikely that prolonged Piwi retention in nuage is an indirect consequence of delayed mitotic exit.

The above live imaging observations indicated that budding off of the Piwi granule from mitotic nuage occurs near the center of dividing cells (S1 Video). Because Klp10A localizes to the central spindle and nuage components likely interact with Klp10A at the central spindle (Fig 3), we next examined how budding of the Piwi granule occurs at the central spindle. Using GFP-Piwi combined with mCherry-α-Tubulin, we found that the GFP-Piwi signal was dynamically associated with the central spindle MTs and moved back and forth along MTs, gradually decreasing in intensity until it completely dissipated (Fig 5C and S3 Video). In contrast, in *klp10A^RNAi* GSCs/SGs, the majority of Piwi granules did not localize to the central spindle, and even when they are associated with the central spindle, Piwi did not exhibit gradual dispersion as observed in the control (Fig 5D, S4 Video).

These results indicate that budding off and release of Piwi from mitotic nuage occurs along the central spindle MTs. We hypothesized that the interaction of nuage with the central spindle MTs facilitates the dissociation of Piwi from nuage. To more directly test this idea that MTs mediate Piwi dissociation from mitotic nuage, we sought to allow cells to exit mitosis in the absence of MTs. To achieve this, we combined *ex vivo* colcemid treatment with a *mad2* mutation. Colcemid treatment effectively depolymerized MTs (S9F Fig)[31], which would normally cause metaphase arrest due to the spindle assembly checkpoint (S9B and S9D Fig) [31,

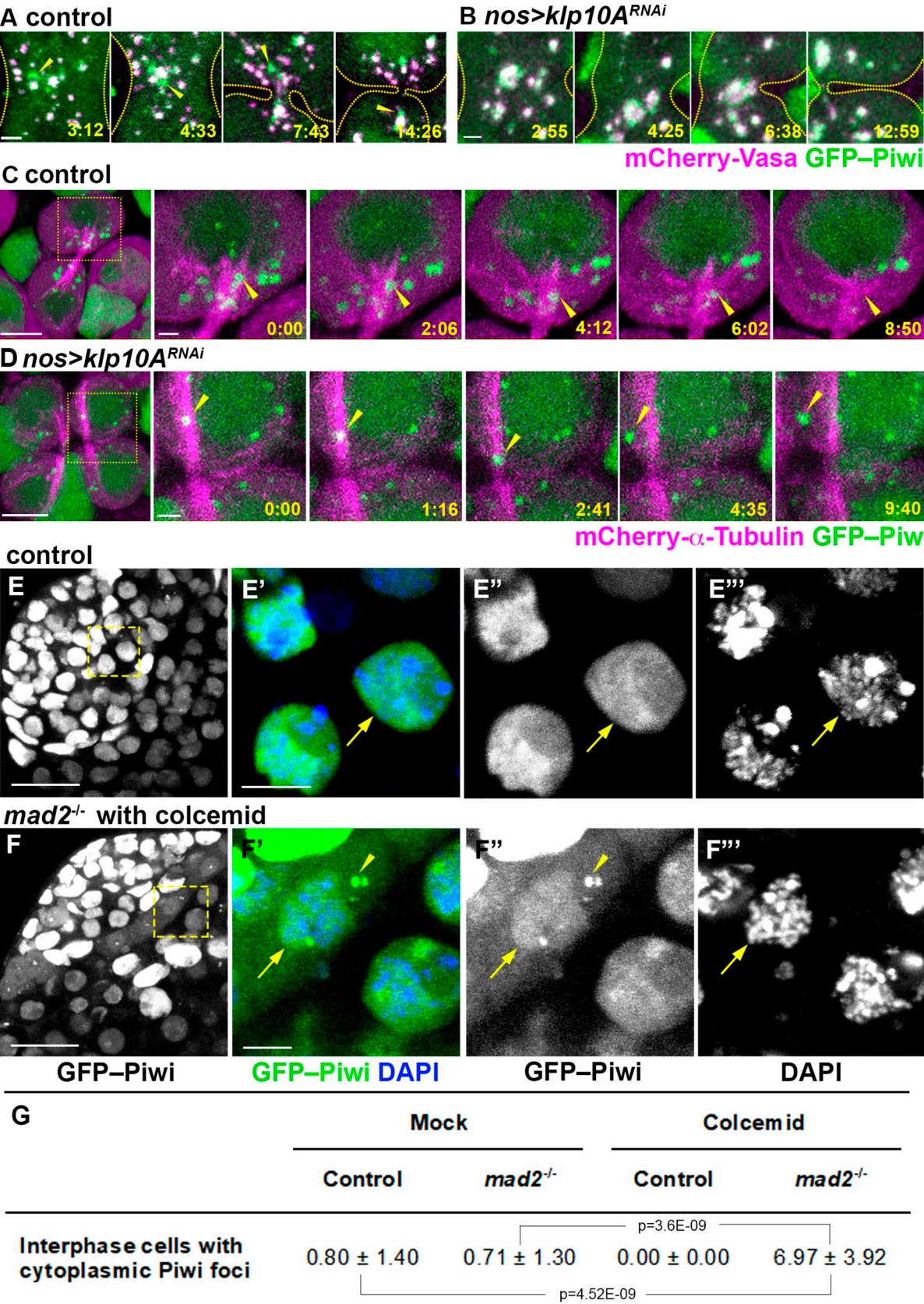

**Fig 5. Piwi dissociates from mitotic nuage at the central spindle in a *klp10A*-dependent manner.** (A-B) Live imaging of mCherry-Vasa (magenta) and GFP-Piwi (green) in the central region of telophase SGs in wild type (A) and in *klp10A^RNAi^* (B) testes. The elapsed time after anaphase-B onset is indicted in min:sec. Arrowheads point to Piwi-GFP signals budding off from the mitotic nuage. Dotted lines indicate the outline of cells. Bars: 1 μm. C-D) Live imaging of mCherry-α-Tubulin (magenta) and GFP-Piwi (green) in telophase SGs in wild type (C) and *klp10A^RNAi^* (D) testes. Squares in the first panels indicate the zoomed regions shown by the later panels. Arrowheads in (C) point to a mitotic nuage particle, initially positive for Piwi and gradually losing the signal. Open arrowheads (D) point to a Piwi-positive nuage particle, sliding along and later releasing the MT bundle, without decreasing GFP-Piwi signal. Bars: 5 μm, and 1 μm for zoomed regions. E-F) GFP-Piwi (green) localization in interphase control SGs (E) or *mad2* null mutant SGs after 3h colcemid treatment (F). Arrowheads point to Piwi-GFP-positive nuage. Arrows point to interphase nuclei. Bars: 20 μm in left panels, 5 μm in insert panels. G) Frequency (cells/per testis) of interphase germ cells with GFP-Piwi positive nuage 3h after *ex vivo* colcemid treatment. *n* = 13–18 testes were scored for each experiment. p values of t-tests are provided.

53]. However, when this is combined with a *mad2* mutation, which inactivates the spindle assembly checkpoint, cells exit mitosis even in the absence of MTs (S13 Fig) (Li et al. 2010). Under this condition, we found that Piwi remained in cytoplasmic nuage after mitotic exit, demonstrating that Piwi's release from mitotic nuage and return to the nucleus is mediated by MTs (Fig 5E–5G).

### *klp10A* is required for depolymerization of central spindle MTs

The above results led us to hypothesize that the central spindle serves as a platform for Piwi dissociation from mitotic nuage to allow Piwi to return to the nucleus. How does Klp10A, which is a kinesin motor that bends and depolymerizes MTs [26], regulate this process? Because central spindle MTs seemed to facilitate dissociation of Piwi from mitotic nuage (Fig 5), and because Klp10A localizes to the central spindle (S7 Fig), we investigated whether Klp10A may regulate the integrity of the central spindle.

To test this idea, we examined the morphology of the central spindle in control vs. *klp10A^RNAi^* testes. First, we found that the frequency of GSCs or SGs that contain central spindle MTs was significantly higher in *klp10A^RNAi^* compared to control testes (Fig 6A–6C), suggesting that Klp10A promotes depolymerization of central spindle MTs. In control germ cells, disassembly of the central spindle and completion of S phase, as assessed by pulse labeling with 5-ethynyl-2'-deoxyuridine (EdU), are nearly in synchrony: most of the S phase cells (EdU+) contain central spindle (Fig 6D, arrowhead), whereas all post-S phase cells (EdU-) have resolved central spindles (Fig 6F). In stark contrast, about half of post-S phase cells still contained the central spindle in *klp10A^RNAi^* testes (Fig 6E, arrow, Fig 6F), indicating that central spindle disassembly is delayed in the absence of *klp10A*. Importantly, increased frequency of germ cells with the central spindle is not due to skewed composition of cell cycle stages in *klp10A^RNAi^* germ cells, because *klp10A^RNAi^* germ cells had comparable frequency of being in S phase (22.6±8.6% EdU$^+$ in control, *n* = 89 GSC vs. 24.6±15.6% EdU + in *klp10A^RNAi^*, n = 103 GSC, n.s.) or in M phase (0.22±0.03 mitotic GSCs/testis in control, n = 160 vs. 0.194±0.01 mitotic GSCs/testis in *klp10A^RNAi^*, n = 222, n.s.).

Taken together, these results show that *klp10A* is required for depolymerization of central spindle MTs. Based on the observation that Piwi dissociates from mitotic nuage along the central spindle as it disassembles, we speculate that Piwi's dissociation from the central spindle is driven by MT depolymerization, which is facilitated by Klp10A, a MT-depolymerizing kinesin. Understanding how depolymerizing MTs can facilitate dissociation of Piwi from mitotic nuage awaits future investigation.

## Discussion

In this study, we reveal an unexpected, dynamic localization of piRNA pathway machinery during the cell cycle of mitotically proliferating germ cells (GSCs and SGs) in the *Drosophila*

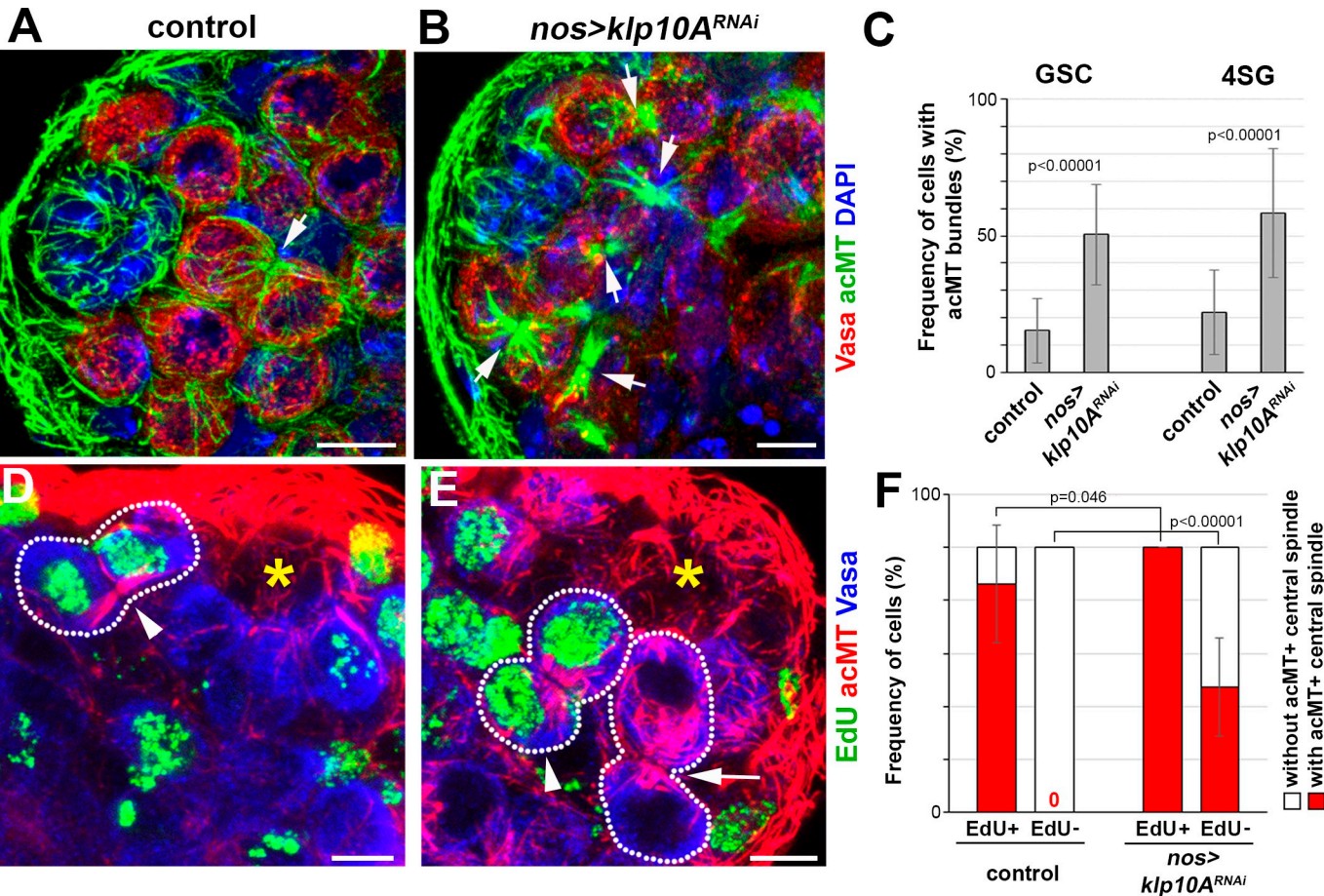

**Fig 6. Klp10A promotes depolymerization of central spindle MTs.** A, B) Apical tip region of wild type (A) and *nos>klp10A^RNAi* (B) testes. Vasa (red), acetylated MTs (acMTs) (green) and DAPI (blue). Arrows indicate acMT bundles of central spindles between interconnected cells. Bars: 5 μm C) Frequency of acMT bundles in GSCs (n = 181 for control, n = 201 for *klp10A^RNAi*) and 4-cell SGs (n = 128 and n = 95). Error bars indicate SD. p values of t-tests are provided. D-E) Apical tip region of a wild type (D) and a *klp10A^RNAi* testis (E) with EdU (green) incorporation after 45min incubation period. Testes were stained for acMTs (red) and Vasa (blue). Arrowheads indicate EdU-positive GSC-GB pairs with acMT bundle. Arrow indicates EdU-negative GSC-GB pairs with acMT bundle in *klp10A^RNAi*. Asterisks indicate the hub, dotted lines indicate GSC-GB pairs. Bars: 5 μm. F) Frequency of acMT bundle in EdU-positive (n = 32 for control, n = 24 for *klp10A^RNAi*) and EdU-negative GSCs (n = 69 for control, n = 79 for *klp10A^RNAi*). Error bars indicate SD. p values of t-tests are provided.

testis. Our study is the first to describe a dynamic compositional change in nuage during the cell cycle. Piwi is nuclear in interphase, but associates with nuage specifically during mitosis and then returns to the nucleus at the end of the mitosis. Our study provides several novel insights into the cell biological aspects of piRNA biogenesis.

Our data suggest that this 'nuage cycle' in mitotic germ cells of the testis is partly facilitated by disassembly of central spindle MTs: Klp10A regulates depolymerization of central spindle MTs during telophase, and Piwi dissociates from nuage at depolymerizing central spindle MTs. How can depolymerizing MTs facilitate dissociation of Piwi from the nuage compartment? Although the underlying molecular mechanisms remain elusive, the physical interaction of piRNA pathway components and MTs have also been documented [54–56]. In addition, MTs have been shown to regulate the assembly, maturation, and disassembly of stress granules, another phase-separated compartment similar to nuage [57–60]. Given these earlier studies, our observation might reflect a generalizable regulatory feature of the piRNA pathway. Recently, it was shown that liquid droplets of Tau protein concentrate tubulin dimers

to facilitate MT formation, and that MTs within the Tau droplets exhibit liquid-like properties [61]. Polymerized MTs within Tau droplets then deforms Tau droplets [61]. Likewise, tubulin-binding proteins within nuage [54–56] may increase tubulin concentration, leading to MT polymerization. It might in turn deform nuage and change biophysical characteristics of phase separation, leading to release of Piwi from nuage. Although our results show the role of Klp10A in dissociating Piwi from nuage at the end of mitotic spindle, underlying mechanism of how Klp10A interacts with piRNA pathway components in facilitating Piwi's dissociation on the depolymerizing MTs remains unknown. Klp10A protein does not have any noticeable domains other than central motor domain, and the structure does not hint as to how Klp10A might interact with piRNA pathway components. It is possible that Klp10A's interaction with nuage components are mediated by unidentified linker protein(s). Also, it remains unknown how Piwi associates with nuage at the mitotic entry, which is likely independent of Klp10A. Nuclear envelope breakdown may be sufficient for Piwi to associate with nuage. Alternatively, there may be additional post-translational regulations that enhance Piwi's affinity to nuage.

We demonstrate that Klp10A is involved in piRNA biogenesis. Loss of *klp10A* leads to a global upregulation of piRNA expression but only modest depletion of a subset of transposons. The failure of *klp10A* knockdown to show depletion of a majority of TEs, despite elevated levels of cognate piRNAs, could be because *klp10A* influences piRNA biogenesis only during limited stages of spermatogenesis, i.e. GSCs and SGs. For example, TE transcripts are not regulated by the piRNA pathway in germ cells of certain stages of spermatogenesis [62], and defects caused by *klp10A* depletion may be masked by unchanged TE expression in other stages of spermatogenesis. Our cytological data show that *klp10A* is required for the release of Piwi from the mitotic nuage, such that Piwi can re-enter the nucleus upon completion of mitosis. As a result of delayed return of Piwi to the nucleus, overall nuclear amount of Piwi was reduced in $klp10A^{RNAi}$. How do these defects result in increased piRNA production? It is possible that the prolonged localization of Piwi to the nuage in $klp10A^{RNAi}$ may enhance the production of pre-piRNAs (or its loading to Piwi), leading to subsequent dysregulation of ping-pong activity to cause upregulation of piRNAs. Alternatively, because Piwi represses the expression of TEs and certain pre-piRNAs [63, 64], reduced nuclear Piwi might lead to derepression of TEs and pre-piRNAs, which, in turn, may result in enhanced piRNA production through intact ping-pong activity in $klp10A^{RNAi}$ SGs.

Expression of piRNAs in male germ cells appears to be distinct from those found in nurse cells during female germline development [65]. Nurse cells are highly polyploid post-mitotic cells and thus do not undergo nuclear envelope breakdown as do SGs [66]. In nurse cells of the *Drosophila* ovary, Piwi needs to be loaded with piRNAs to translocate into the nucleus [6, 67, 68]. However, Piwi localization to nuage is not visible in wild type cells, likely because Piwi only transiently associates with nuage prior to nuclear translocation. Only when the piRNA pathway is completely compromised, as in the *aub ago3* double mutant, can the failure of Piwi to enter the nucleus be detected, leading to its accumulation in nuage [6]. In contrast to this mechanism of piRNA loading and nuclear translocation of Piwi in post-mitotic nurse cells, our finding suggests that mitotically dividing germ cells (SGs) in the testis may utilize mitosis (nuclear envelope breakdown) as a means to load Piwi with piRNAs. Nuclear envelope breakdown may represent a robust and efficient way of loading Piwi with piRNAs in SGs that divide every ~12 hours.

Taken together, we propose that Klp10A-dependent MT depolymerization at the central spindle facilitates Piwi dissociation from nuage to promote its translocation back into the nucleus. Piwi's interaction with nuage during mitosis might impact piRNA biogenesis, as indicated by piRNA profiling. In summary, the present study reveals a novel cell biological

mechanism by which nuclear and cytoplasmic arms of piRNA biogenesis communicate during the cell cycle.

## Materials and methods

### Fly husbandry and transgenic flies

Flies were raised in standard Bloomington medium at 25˚C. The following stocks were used. *UAS-gfp-klp10A* [69], *gfp-vas* [70], *gfp-piwi* (a gift from Katalin Fejes Tóth) [15], *mCherry-vas* (a gift from Elizabeth Gavis)[71], and the following stocks, obtained from the Bloomington Stock Center: *nos-gal4* [72], *UAS-upd* [73], *UAS-gfp* (Spana E., 1999, personal communication to FlyBase, ID: FBrf0111645), *UAS-gfp-aub* [74], *UAS- mCherry-α-tubulin* [75], *UAS-klp10A*$^{TRiP.HMS00920}$ (Flybase: FBrf0214641, FBrf0212437), *mad2*$^{EY21687}$ [53], Df (3L)BSC437 (FlyBase: FBrf0204472)

### Immunoprecipitation and western blotting

For immunoprecipitation, 150 pairs of testes were lysed in 0.5 ml HEPES based buffer [50mM HEPES pH 7.5, 250mM NaCl, 0.1% Nonidet P40, 0.2% Triton X-100, supplied with cOmplete protease inhibitor mixture (Roche)]. In Klp10A co-immunoprecipitation (co-IP) experiments, the cleared lysates were incubated with rabbit anti-Klp10A serum (1:200, [37]) for 4 hours, extended by 2 extra hours after supplement with Protein A Dynabeads (Life Technologies). The beads were washed four times with lysis buffer and proteins were eluted with SDS-PAGE protein sample buffer. For co-IP experiments with GFP tagged proteins, the cleared lysates were incubated with GFP-Trap magnetic beads (Chromotek) according to manufacturer's instructions. To detect precipitated proteins on western-blots, the following primary antibodies were combined with horseradish peroxidase conjugated secondary antibodies (Abcam): guinea-pig anti-Klp10A (1:10,000, [37]), rabbit anti-GFP (1:3000, ab290, Abcam), rabbit anti-Vasa d-26 (1:3000, Santa Cruz Biotechnology), rabbit anti-Piwi (1:3000 [76]), rabbit anti-Aub (1:3000 [76]), and mouse anti-Ago3 (1:3000 [12]).

### Mass-spectrometry

Klp10A was purified either with anti-Klp10A or anti-GFP nanobodies from GSC-enriched testes (250 pairs of *nos-gal4>UAS-upd* testes or 1500 pairs of *nos-gal4>UAS-upd, UAS-gfp-klp10A* testes, respectively) were dissected, and homogenized in lysis buffer (10 mM Tris-HCl pH 7.5; 150 mM NaCl; 0.5 mM EDTA, 0.1% NP40 and protease inhibitor cocktail (cOmplete, Roche)) for 30 min at 4˚C. After centrifugation at 13,000 rpm for 10–15 min at 4˚C the supernatant was saved as whole cell extract. The protein concentration was measured by absorbance at 562 nm using Pierce™ BCA Protein Assay Kit (Thermo Fisher Scientific). The whole cell extract of 1 μg total protein mass was incubated with 40 μl packed guinea-pig anti-Klp10A [37] conjugated Protein A-Dynabeads (Thermo Fisher Scientific) for 2h at 4˚C, or with 25 μl packed GFP-Trap agarose beads (Chromotek) for 3 hours at 4˚C. Beads were washed three times with wash buffer (10 mM Tris-HCl pH 7.5; 150 mM NaCl; 0.5 mM EDTA), and the proteins were eluted in LDS sample loading buffer (1.5x) at 100˚C for 5 min and separated on a 10% Bis-Tris Novex mini-gel (Invitrogen) using the MES buffer system. In the anti-Klp10A pull down experiments the gel was stained with Coomassie dye and excised into ten equally sized segments. These segments were analyzed by LC/MS/MS (MS Bioworks, Ann Arbor, MI). The gel digests were analyzed by nano LC/MS/MS with a Waters NanoAcquity HPLC system interfaced to a Thermo Fisher Q Exactive. Peptides were loaded on a trapping column and eluted over a 75 μm analytical column at 350 nL/min; both columns were packed with Luna

C18 resin (Phenomenex). The mass spectrometer was operated in data-dependent mode, with MS and MS/MS performed in the Orbitrap at 70,000 FWHM resolution and 17,500 FWHM resolution, respectively. In the anti-GFP pull down experiments after electrophoresis the gel was stained with coomassie and excised into two equally sized segments. The gel digests were analyzed by LC-MS/MS with LTQ Orbitrap XL mass spectrometer (Thermo Fisher Scientific) at Fred Hutchinson Cancer Research Center, Proteomics Resource.

## Immunofluorescence staining and microscopy

Fixation and immunofluorescence staining of testes was performed as described previously [37]. The following primary antibodies were used. Mouse anti-Fasciclin III (FasIII; 1:100; 7G10, developed by Goodman C, obtained from Developmental Studies Hybridoma Bank; [77]), rabbit anti-Vasa (1:200; d-26, Santa Cruz Biotechnology), mouse anti-α-tubulin (4.3; 1:50; developed by C. Walsh [78] and obtained from the Developmental Studies Hybridoma Bank), mouse anti-acetylated-Tubulin (1:100, 6-11B-1, Sigma), rabbit anti-phosphorylated (Thr3) histone H3 (PH3) (1:200; clone JY325, Upstate),mouse anti-LaminB (1:200; C20; Santa Cruz Biotechnology), guinea pig anti-Traffic jam (Tj; 1:400; a gift from Dorothea Godt; [79]), rabbit and guinea-pig anti-Klp10A (1:3000/1:1000 respectively; [37]). Alexa Fluor-conjugated secondary antibodies were used (1:200; Thermo Fisher Scientific), EdU was detected by Click-iT Plus EdU Imaging Kit with Alexa Fluor 647 (Thermofisher). Testes were mounted into VECTASHIELD media with 4',6-diamidino-2-phenylindole (DAPI; Vector Labs). Images were captured using a Leica TCS SP8 confocal microscope with a 63×oil-immersion objective (NA = 1.4) and processed by ImageJ software [80]. Colocalization of nuage components with Klp10A or acMTs were quantified by calculating Mander's coefficient [81, 82] in ImageJ.

## *Ex vivo* treatment of *Drosophila* testis

To enrich cells in metaphase testes were dissected and transferred to Schneider's insect medium (Sigma) containing MG132 (20 μM final concentration, Sigma-Aldrich) [52] or colcemid (100μM final concentration, Calbiochem) [31]. After 4.5h incubation with MG132 or colcemid at 25˚C, GSCs/SGs arrested in metaphase with or without intact bipolar mitotic spindles, respectively. For EdU incorporation the dissected testes were incubated for 45min (10 μM, Thermofisher) in Schneider's insect medium (Gibco) at 25˚C.

## RNA extraction, library preparations, and sequencing

Total RNAs were extracted by Trizol. 300 pairs of testes were used for each sample, and biological triplicates for each genotype were used. For small RNA sequencing, 2.5 μg of total RNAs were treated with DNase I (Amplification Grade, ThermoFisher) and recovered by RNA Clean & Concentrator-5 columns (Zymo). Libraries were generated from DNaseI treated RNAs using reagents from the TruSeq Small RNA Library Prep Kit (Illumina), while following a previously published protocol from [83] that uses a terminator oligo to deplete *Drosophila* 2S rRNA from the final sequencing libraries. The libraries were sequenced in one lane on a HiSeq 2500 machine (Illumina) at the UCLA Broad Stem Cell Research Center BioSequencing Core. The terminator oligo complimentary to Drosophila 2S rRNA was TAC AAC CCT CAA CCA TAT GTA GTC CAA GCA /3SpC3/ (Integrated DNA Technologies).

For mRNA sequencing, 2.5 μg of total RNAs were treated with DNase I (Amplification Grade, ThermoFisher) and recovered by RNA Clean & Concentrator-25 columns (Zymo). DNaseI treated RNAs were then treated with Ribo-Zero Gold rRNA Removal Kit (Illumina). Libraries were generated from rRNA depleted RNAs using TruSeq Stranded Total RNA

Sample Prep Kit (Illumina). The libraries were sequenced by a HiSeq 4000 machine (Illumina) at the UCLA Broad Stem Cell Research Center BioSequencing Core.

## RNA-seq data analysis

Three biological replicates of each condition were used to test for statistical significance and comparative analysis. Raw qseq files were converted to fastq format using custom Python and awk scripts. Small RNAs were clipped from 3' adapter sequences using Trimmomatic 0.27 [84]. Reads mapping with at most 1 mismatch to rRNA and miRNA hairpin sequences were parsed out by Bowtie2 v2.2.3 [85]. The remaining reads were aligned to the *D. melanogaster* transcriptome (Dm3) with at most 1 mismatch allowed using Bowtie2 v2.2.3. Abundance estimation was done at annotated genes, piRNA clusters, and TEs using the eXpress pipeline within the piPipes suite [86, 87]. We used Flybase for annotation of protein coding genes and TEs. piRNA cluster annotations were done according to [1]. For piRNA analysis, we selected reads that were 23–29 nucleotides in length to ensure other endo-siRNA species did not influence downstream analysis. miRNA hairpin reads were used to normalize between libraries. Differential expression analysis was done in DESeq2 3.7 using the Wald test and adjusted p-values were corrected using the Benjamin-Hochberg procedure with an FDR threshold of 0.01 [88]. Plots were generated using R. Sequencing data is available at GEO: accession number: GSE122596.

**Ping-pong activity analysis.**   piRNA reads were mapped directly to the *D. melanogaster* transcriptome (Dm3) using Bowtie2 v2.2.3 [85]. BAM files from Bowtie2 were converted to BED files using several pipelines, including BEDTools 2.7 from the piPipes suite, which assigns nucleotide positions of each read across TE transcript bodies [86, 89]. In addition to nucleotide position, reads were categorized as sense and antisense. Sense was defined as piRNAs that were derived from cleavage of the TE mRNAs and antisense was defined as piRNAs that were antisense to annotated TE transcripts. Custom Python scripts were used to calculate the number of 5' to 5' complementarity between sense and antisense reads. Ping-pong ratios were calculated for each transposon feature across all libraries. To calculate the ping-pong ratio of each transposon, we used custom Python scripts to calculate the number of piRNAs in which sense piRNAs with an A at the 10-nt position or antisense piRNAs with a U at the 1-nt position showed 10 nt of complementarity from the 5' end. We then divided the number of such pairs by the total number of piRNA reads. The resulting ratio allowed for quantification of ping-pong activity, without needing to normalize for library size. The calculation of ping-pong ratio was done using custom Python scripts. Plots were generated using R.

**mRNA-seq data analysis.**   Raw qseq files were converted to fastq format using custom Python and awk scripts. Due to the repetitive nature of transposon sequences, reads were aligned to the *D. melanogaster* genome (Dm3) by STAR-2.6.0a while allowing for up to 100 multi-alignments per read [90]. Feature and abundance estimation were determined using TEtranscripts 2.0.3 [91]. TEtranscripts initially distributes multi-mapped reads evenly among potential matches and optimizes the distribution of multi-mapped reads using the Expectation Maximization approach. Variance was measured based on normalized gene expression counts and we performed principal components analysis and found samples derived from wild-type clustered separately from *klp10A^RNAi* on Principle Component 1 (S6B Fig). Differential analysis was done using DESeq2 3.7 using the Wald test and p-values were corrected using the Benjamin-Hochberg procedure with an FDR threshold of 0.01 [88]. All Plots were generated using R. Pipeline of Analysis is described in (S1A and S1B Fig).

## Live imaging

Testes from newly enclosed flies were dissected in Schneider's Drosophila medium (Gibco) and prepared for live imaging as described previously [92]. The testis tips were placed into a drop of medium in a glass-bottom chamber and were covered by regenerated cellulose membrane (Spectrum Lab). The chamber was mounted on a three-axis computer-controlled piezo-electric stage. An inverted Leica TCS SP8 confocal microscope with a 63× oil immersion objective (NA = 1.4) was used for imaging. Live imaging was performed at ambient room temperature. Images were processed using ImageJ software.

## Supporting information

**S1 Fig. Pipelines of small RNA and mRNA sequence analyses.** A) Pipeline of small RNA sequence analysis. B) Pipeline of mRNA sequence analysis.
(TIF)

**S2 Fig. Distribution of 5' nucleotide identity for reads of 23–29 nucleotides vs. 20–22 nucleotides in length in control and *klp10A^RNAi* testes.**
(TIF)

**S3 Fig. Analysis of small RNAs in *klp10A^RNAi* testes.** A-B) Pairwise comparisons of global miRNA hairpin abundance (A) and piRNA expression (B) between biological replicates. (C) difference in reads in different small RNA classes. Reads were normalized by total library reads. piRNA reads (23–29 nt) are increased in *klp10ARNAi* vs. wild-type, whereas siRNA (20–22 nt) or miRNA (reads mapping to miRNA hairpin features) reads are not.
(TIF)

**S4 Fig. Characterization of piRNA density across TE transcripts.** A-B) Density of sequenced piRNAs (blue: sense; red: antisense) across *FB-element and BARI1*.
(TIF)

**S5 Fig. Ping-pong pathway activity remain unaltered after *klp10A* depletion.** A) Histogram showing the distribution of antisense and sense piRNA pairs of piRNAs mapping to transposons. B) The box-plots show the distribution of ping-pong ratios of each transposon. Each box-plot is a different biological replicate. The Ping-pong ratio of each transposon was calculated by taking the sum of piRNA reads in which sense piRNAs with a 10 nt A and antisense piRNAs with a 1nt U showing 10 nucleotide complementarity from the 5' end and dividing it with the total number of piRNA reads.
(TIF)

**S6 Fig. Characterization of RNAseq datasets.** A) Total library reads for each RNAseq library B) Principle component analysis of wild-type (n = 3 replicates) and *klp10A^RNAi* (n = 3 replicates) RNAseq libraries. C) Scatter plot showing mean genic abundance of *klp10A^RNAi* versus wild-type libraries.
(TIF)

**S7 Fig. Klp10A localization at the central spindle of GSCs/SGs.** Localization of acetylated MTs (acMTs) (red), Klp10A (green), and DNA (blue) in the apical region of a wild type testis (A), and in a telophase GSC-GB pair of a wild type testis (B). Arrows point to central spindle. Bars: 5 μm.
(TIF)

**S8 Fig. Identification of cell cycle stage for analysis of Piwi-Vasa colocalization.** A-C) Same images as Fig 4A–4C and 4D–4F) same images as Fig 4E–4G are shown with additional α-

Tubulin (blue) and DAPI (gray) channels to precisely define their cell cycle stages. Cytoplasmic α-Tubulin staining (without MT bundles of central spindle MTs) combined with decondensed DAPI staining indicate cells in G2 phase (A, D). Spindle α-Tubulin staining and condensed chromosomes indicate metaphase (B, E). Remnant of central spindle (by α-Tubulin staining) and decondensed chromosome indicate G1 phase (or S phase) of the cell cycle (completion of telophase) (C, F).
(TIF)

**S9 Fig. Piwi-Vasa colocalization in mitotic GSCs/SGs.** A-D) efficiency of mitotic arrest by colcemid or MG132. Apical tip of testes after 4.5h *ex vivo* mock (A), colcemid (B), or MG132 treatment (C). PH3 (green), DAPI (white). Bars: 20 μm. D) Number of mitotic cells per testis after 4.5h colcemid or MG132 treatment. Error bars indicate SD. P-values of t-tests are provided. E-G) Mitotic SGs after mock (E), colcemid (F), or MG132 (G) treatment. Colcemid efficiently depolymerizes MTs, whereas MG132 arrest cells in mitosis with intact spindle. α-Tubulin (cyan), DAPI (yellow). Bars: 5 μm. H-J) GFP-Piwi (green) and mCherry-Vasa (red) localization in SGs after 1h *ex vivo* culture with mock (H), colcemid (I) or MG132 (J) treatment. Magnified images of mitotic cells in H-I (boxed) are shown in H'-J". Mitosis can be judged based on the lack of perinuclear Vasa localization and the lack of nuclear Piwi localization. Arrowheads point to mitotic nuage with Piwi-Vasa colocalization. Bars: 5 μm.
(TIF)

**S10 Fig. GFP-Piwi localization changes during mitotic exit in *klp10A^RNAi* GSCs/SGs.** A) GFP-Piwi (green) localization during mitosis in a control or *klp10A^RNAi* germ cells. Mitotic cells are encircled by dotted lines. Time in minutes. Bar: 5 μm. B) Quantification of GFP-Piwi localization during the mitotic exit of GSCs and SGs.
(TIF)

**S11 Fig. Piwi stays in nuage after mitotic exit in *klp10A^RNAi* germ cells.** A) GFP-Piwi is nuclear in interphase GSCs/SGs in control testes. B) GFP-Piwi colocalizes with Vasa at the nuage of interphase GSCs/SGs in *klp10A^RNAi* germ cells. Cytoplasmic Vasa and α-Tubulin staining as well as DAPI staining indicates that these cells are in interphase. GFP-Piwi (green), Vasa (magenta). Arrowhead points to nuage-localized Piwi in interphase *klp10A^RNAi* GSCs/SGs. Bars 5μm. C) Number of interphase GSCs/SGs with nuage-localized Piwi per testis. n = 30 testes per genotype. p value of t-tests is provided.
(TIF)

**S12 Fig. Nuclear Piwi level is decreased in *klp10A^RNAi* germ cells.** A) GFP-Piwi (green) in the apical tip of control and *klp10A^RNAi* testes. Tj (red) identifies cyst stem cells and early cyst cells. DAPI (blue). The area surrounded by dotted lines with asterisks indicates hub. GSC nuclei are encircled by green line, hub cell nuclei by magenta line. Bars: 5 μm. B) Photon counts in nuclear areas of hub cells (magenta) and GSCs (green) in *klp10A^RNAi* testes. Each data point represents individual single nucleus. Error bars indicate SD, p-values from t-tests are provided. As *klp10A* is knocked down only in germline (with *nos-gal4*), photon counts in hub cell nucleus is expected to be unchanged, and served as an internal control. Piwi nuclear level was reduced in GSCs upon knockdown of *klp10A*. Note that similar reduction in the nuclear Piwi was observed in SGs as well: however, hub cells and GSCs were used to quantify Piwi levels, because their juxtaposition allows accurate comparison.
(TIF)

**S13 Fig. *mad2* mutant SGs exit mitosis in the absence of MTs due to defective spindle assembly checkpoint.** A) *mad2* mutant SGs do not arrest in mitosis after clocemid-induced

MT depolymerization. B) After prolonged MT depolymerization with colcemid, some *mad2* mutant SGs exit mitosis: their nuclei are slightly larger because they are tetraploid, having exited mitosis without chromosome segregation (yellow dotted line indicates SGs that exited mitosis). Vasa (green), PH3 (red), Lamin-B and FasIII (blue). Bars: 5 μm.
(TIF)

**S1 Video. Dynamics and composition of mitotic nuage in wild type SGs.** Dynamics of GFP-Piwi (green) and mCherry-Vasa (red) in mitosis of wild type SGs. Time 0 is set to ana-phase B onset. Z-stacks were collected with the indicated time intervals with Z-step size 0.5 μm and 9 steps per time point. Bar on first frame: 5 μm. Note that at the end of mitosis, cyto-plasmic nuage is Vasa+, Piwi-.
(AVI)

**S2 Video. Dynamics and composition of mitotic nuage in a SG upon knockdown of *klp10A*.** Dynamics of GFP-Piwi (green) and mCherry-Vasa (red) in mitosis of *klp10A^RNAi* GSC (left bottom) and SG (upper right). Time 0 is set to anaphase B onset. Z-stacks were col-lected with the indicated time intervals with Z-step size 0.5 μm and 11 steps per time point. Bar on first frame: 5 μm. Note that at the end of mitosis, cytoplasmic nuage is Vasa+, Piwi+, although some portion of Piwi returns to the nucleus.
(AVI)

**S3 Video. Piwi disassembles along the central spindle MTs in a wild type GSC during mitotic exit.** Lime-lapse live imaging of GFP-Piwi (green) and mCherry-α-Tubulin (red) in a wild type GSC. Time 0 is set to first frame (0 min) in anaphase of the cell cylce. Z-stacks were collected with the indicated time intervals with Z-step size 0.5 μm and 8–12 steps per time point (note that when Piwi+ somatic nuclei move into the focal plane, those frames were man-ually removed). Bar on first frame: 5 μm.
(AVI)

**S4 Video. Defective Piwi disassembly in late telophase/G1 SGs upon knockdown of *klp10A*.** Lime-lapse live imaging of GFP-Piwi (green) and mCherry-α-Tubulin (red) in *klp10A^RNAi* GSCs. Time 0 is set to first frame (0 min) in late telophase/G1 phase of the cell cycle. Z-stacks were collected with the indicated time intervals with Z-step size 0.5 μm and 6–10 steps per time point (note that when Piwi+ somatic nuclei move into the focal plane, those frames were manually removed). Bar on first frame: 5 μm.
(AVI)

**S1 Table. Results of anti-GFP pull downs from *nos>gfp, upd* vs. *nos>upd, gfp-klp10A* tes-tes.** piRNA pathway proteins are highlighted by bold orange text, proteins of GO categories: pole plasm assembly, gene silencing by RNA, posttranscriptional gene silencing, negative regu-lation of transposition, P granule are highlighted by blue cell, proteins of GO categories: microtubule, microtubule-associated complex, microtubule-organizing center, spindle are highlighted by boxes. Additional table columns of sheets provide a full list of Drosophila pro-teins of the above.
(XLSX)

**S2 Table. Results of preimmune IgG vs. anti-Klp10A pull downs from *nos>upd* testes.** piRNA pathway proteins are highlighted by bold orange text, proteins of GO categories: pole plasm assembly, gene silencing by RNA, posttranscriptional gene silencing, negative regulation of transposition, P granule are highlighted by blue cell, proteins of GO categories: microtubule, microtubule-associated complex, microtubule-organizing center, spindle are highlighted by boxes. Additional table columns of sheets provide a full list of Drosophila proteins of the

above.
(XLSX)

**S3 Table. Results of repeat experiment of preimmune IgG vs. anti-Klp10A pull downs from *nos*>*upd* testes.** Control experiment is the same between Source Data 2 and Source Data 3. piRNA pathway proteins are highlighted by bold orange text, proteins of GO categories: pole plasm assembly, gene silencing by RNA, posttranscriptional gene silencing, negative regulation of transposition, P granule are highlighted by blue cell, proteins of GO categories: microtubule, microtubule-associated complex, microtubule-organizing center, spindle are highlighted by boxes. Additional table columns of sheets provide a full list of Drosophila proteins of the above.
(XLSX)

**S4 Table. Results of anti-Klp10A pull downs from *nos*>*dpp* testes.** Identified proteins are ranked based on peptide score. piRNA pathway proteins are highlighted by bold orange text, proteins of GO categories: pole plasm assembly, gene silencing by RNA, posttranscriptional gene silencing, negative regulation of transposition, P granule are highlighted by blue cell, proteins of GO categories: microtubule, microtubule-associated complex, microtubule-organizing center, spindle are highlighted by boxes. Additional table columns of sheets provide a full list of Drosophila proteins of the above.
(XLSX)

**S5 Table. Results of repeat experiment of anti-Klp10A pull downs from *nos*>*dpp* testes.** piRNA pathway proteins are highlighted by bold orange text, proteins of GO categories: pole plasm assembly, gene silencing by RNA, posttranscriptional gene silencing, negative regulation of transposition, P granule are highlighted by blue cell, proteins of GO categories: microtubule, microtubule-associated complex, microtubule-organizing center, spindle are highlighted by boxes. Additional table columns of sheets provide a full list of Drosophila proteins of the above.
(XLSX)

**S1 Dataset. Numerical data associated with each graph.**
(XLSX)

## Acknowledgments

We thank Elizabeth Gavis, Katalin Fejes Tóth, Dorothea Godt, Mikiko Siomi, Phil Zamore, Bill Theurkauf, the Bloomington Stock Center and the Developmental Studies Hybridoma Bank for reagents, the Yamashita lab members, Rebecca Tay, Margaret Starostik, Nelson Lau, Sue Hammoud and Lei Lei for discussions and comments on the manuscript.

## Author Contributions

**Conceptualization:** Zsolt G. Venkei, John K. Kim, Yukiko M. Yamashita.

**Data curation:** Charlotte P. Choi, John K. Kim.

**Formal analysis:** Zsolt G. Venkei, Charlotte P. Choi.

**Funding acquisition:** Steven E. Jacobsen, John K. Kim.

**Investigation:** Zsolt G. Venkei, Charlotte P. Choi, Suhua Feng, Cuie Chen.

**Methodology:** Zsolt G. Venkei, Charlotte P. Choi, John K. Kim, Yukiko M. Yamashita.

**Supervision:** John K. Kim.

**Writing – original draft:** Zsolt G. Venkei, Yukiko M. Yamashita.

**Writing – review & editing:** Zsolt G. Venkei, John K. Kim, Yukiko M. Yamashita.

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
