## [Decision Letter · Decision Letter 0]

7 Jan 2020

Dear Dr Yamashita,

Thank you very much for submitting your Research Article entitled 'A kinesin Klp10A mediates cell cycle-dependent shuttling of Piwi between nucleus and nuage.' to PLOS Genetics. Your manuscript was fully evaluated at the editorial level and by independent peer reviewers. The reviewers appreciated the attention to an important topic but identified some aspects of the manuscript that should be improved. Although both reviewers were generally positive, both also raised a number of important points that need to be addressed.  

We therefore ask you to modify the manuscript according to the review recommendations before we can consider your manuscript for acceptance. You may not need to add any additional or new data in order to address the reviewers' concerns, but if you choose to do so then reviewers will be asked to fully critique all new data. Please make sure that your revisions address all the specific points made by each reviewer. In particular, I would draw your attention to reviewer #1 comments regarding data presented in Figures 3&4.

[LINK]

Yours sincerely,

Giovanni Bosco, Ph.D.

Associate Editor

PLOS Genetics

Gregory P. Copenhaver

Editor-in-Chief

PLOS Genetics

Reviewer's Responses to Questions

**Comments to the Authors:**

Reviewer #1: Piwi is an essential protein in metazoan germline maintenance and plays a major role in piRNA-mediated regulation of transposable elements. The mechanistic details of this important biology however remain unresolved. Understanding of Piwi dynamics during cell cycle adds to the mechanistic understanding and will contribute to the elucidation of the mechanism of Piwi function.

This paper describes the relocalization of Piwi from the nucleus to nuage during mitosis. The authors find that Piwi separates from nuage late in mitosis to form granules that appear to associate with the central spindle and migrate along these microtubules to segregate between the dividing cells. The microtubule depolymerising protein Klp10A is required for the latter parts of this process. During the late phase of mitosis it is also associated with the central spindle, and interacts (possibly indirectly) with several nuage components (Aub, Vasa), and somewhat with Piwi. Furthermore, depletion of Klp10A affects piRNA biogenesis, although without major effect on TE silencing.

The story has interesting bones, but there seem to be quite some loose ends and discrepancies that make it difficult to fully evaluate. The data clearly support the movement of Klp10A from centrosome to central spindle and back, as well as the movement of Piwi from nucleus to nuage and back during cell cycle. However the connection between the two, as well as the functional relevance remains more unclear.

The authors show a genetic interaction between the movement of Piwi and klp10A, as it is also clear that the return to the nucleus is delayed in a Klp10A- background. However, this background leads to obvious mitotic defects as the central spindle remains, and the final stages of mitosis look disrupted. It is therefore difficult to determine whether the Piwi phenotype is a direct result of the Klp10A loss or a more indirect effect. The Mass Spec data is interpreted to support a direct role, but in fact Klp10A binds hundreds of proteins making it hard to know whether the finding that Piwi is one of them is meaningful. Furthermore the Mass Spec recovers Piwi in one experiment, but not in another, suggesting that this is not a stable interaction - as would be expected form the cell biology, but makes the interpretation harder. In particular, as soon as some RNA-binding proteins are part of the complex, RNAs and other RNA-binding proteins are likely to associate as well to some degree. Therefore the IP experiments should be repeated in the presence of RNase to give relevance to this interaction.

A second loose end is that it is unclear what the functional relevance of the delayed relocalization of Piwi is. There is a mild effect on piRNA levels, although it is unclear how normalisation has been done to arrive at this conclusion. I didn't find any description of spikes in the sequencing data, and therefore expansion of one class of small RNA might actually be a reduction in other classes. This notwithstanding, the effect on transposon expression is sporadic and very mild.

Overall, I think there are some interesting observations that are of interest to the wider community. Focussing the story on the relocalization of Piwi and reducing the emphasis on the interaction with Klp10A could make it more cohesive. This however would require a better characterisation of the functionality of the process.

Specific comments:

Mass Spec and coIP data. (Fig1, Table1, Fig 4)

There is a quite remarkable discrepancy between Mass Spec data and Western: Western shows solid co-IP of Ago3 and very faint co-IP of Piwi. Mass Spec barely shows Ago3, but shows more solid Piwi.

Is there any explanation for this effect?

From Mass Spec it appears that Klp10A binds hundreds of proteins. It still binds nuage as well, but this could be very indirect. Once you start pulling on RNA, you might get all kinds of RNA-associated proteins.

Is the interaction RNA-dependent?

The same goes for the coIP data in Fig 4:

The coIP data of Piwi and Vasa in Klp10A RNAi is intriguing, but this could again be just pulling on RNA and getting all kinds of other RNA-binding proteins in the same compartment.

Microscopy data. (Fig 3-6 and videos)

The microscope images are interesting, but it is hard to judge from them how representative the described relocalizations are, and how tightly temporally controlled the relocalization is even in the controls. It is noticeable that in images that don't focus on a protein, the localisation appears less discrete (e.g. Fig 3 Sup1 Vasa at telophase is not clearly on central spindle.)

There also appears to be discrepancy in the staging between Fig 3 and Fig 4: in Fig 3 the point is that Piwi is close to the central spindle at telophase, but in Fig 4 it is already nuclear at telophase and definitely not on the central spindle. Furthermore, the localisation of Piwi to nuage depends much on seeing the right plane. As no nuage is in focus in the control panel it is impossible to tell if Piwi would have been in it or not. It would require full stacks or all east a plane with nuage in it to judge the difference.

The budding off of the PIwi granule is intriguing, but very difficult to see. From the videos it appears that the overall Choreography of nuage is very different in Klp10A as it stays more peripheral. In the control, nuage does go to the region of the central spindle, but the association with central spindle itself seems at least incomplete.

Further, in Klp10A RNAi Piwi localisation seems disorganised in general. Levels are lower, staining is patchy in some, and is overexposed in other images. This makes it difficult to judge how specific the relocalization of Piwi is in Klp10A-. In Fig4 Sup3 the DAPI signal even looks a bit weird and patchy in Klp10A-, suggesting there may be very general missegregation happening. Similarly, Fig 4 Sup 3 images clearly show a delay in the return of Piwi from nuage to nucleus. But is there a defined nucleus already? What does the nuclear membrane look like in klp10A-? Fig 6 actually also suggests that general segregation is altered as the central spindle persists after exit from mitosis, and videos 3 and 4 either are not from comparable stages, or show such severe segregation defect that none of the processes are recognisable in the mutant background.

Together this makes it difficult to judge how specific the described defect is.

Functional data (Fig 2)

It appears from the data that the ping-pong pathway is more active when Piwi remains in nuage longer. However, it is unclear how normalization is done. Could also be that miRNAs/tRNAs/rRNAs are down?

Only a few TEs are affected (and only mildly). Is there a correlation between the affected TEs and the hyperactive ping-pong piRNAs?

Overall this data is still puzzling as it would require increased piRNA cluster transcription as well as increased TE transcription to get higher levels of both sense and antisense piRNAs. Also, the ping-pong pathway is thought to mostly be formed by Ago3 and Aub, so how its overactivity is related to mislocalisation of Piwi is unclear.

Reviewer #2: Yamashita and colleagues identify that Klp10A, a microtubule-depolymerizing kinesin motor protein, is required for proper shuttling of Piwi between the nucleus and cytoplasm, in what are presumed to be nuage, during mitosis. In the absence of Klp10A, piRNA levels are elevated and a small subset of transposons is downregulated, although the reason for this is not revealed in this study. Overall the results are interesting and could have important implications fur further studies exploring nuage and piRNA dynamics.

The manuscript seems to have already undergone multiple rounds of revisions and I did not identify any major issues. A general concern, that I suspect may have derailed it at eLife, is the lack of solid mechanistic insight and a rather fuzzy link between microtubules and nuage and the piRNA machinery. Nonetheless, the results will likely be of broad interest and provide a valuable framework for future studies.

Minor Points

Lines 132-133: "To enrich for GSCs, the self-renewal factor Unpaired (Upd) (Kiger et al., 2001; Tulina and Matunis, 2001) was expressed (nos-gal4>UAS-upd)." A bit more explanation would help here.

Lines 147-150: "Similar interactions between Klp10A and piRNA pathway components (Vasa, Piwi, Aub, Ago3) were observed when extracts from SG tumor was used (nos>dpp, (Kawase et al., 2004; Schulz et al., 2004; Shivdasani and Ingham, 2003))." The rationale for using SG tumor should be noted.

Lines 166-167: "We defined piRNAs as reads of 23-29 nucleotides (nt) in length that did not map to microRNAs or ribosomal RNAs." What is the evidence that these criteria are sufficient to define piRNAs? Perhaps it would make more sense to focus specifically on the known piRNA clusters.

Figure 2A. Error bars are needed.

Figure 2D. Of the transposons that are downregulated, how many have elevated levels of piRNAs? I'm trying to get a sense for whether there is a trend in piRNA upregulation and transposon downregulation, aside from the Hobo and FB examples shown. This should be noted in the results.

Figure 2-Figure 2 supplement 2B. Only piRNA features upregulated or downregualted >1.5 fold are shown, whereas all miRNAs are shown, which makes comparing the two sets of plots rather uninformative, particularly since miRNAs are not color coded similarly to the piRNAs.

Lines 217-218: "When cells enter mitosis, nuage became somewhat larger in size, which we refer to as 'mitotic nuage' hereafter (Figure 3C, G and K)." What defines these foci as nuage?

Lines 238-241. Why does the localization of Piwi during telophase differ so much between figures 3 and 4A?

Piwi is just one of several piRNA factors identified as interactors of Klp10A, yet only Piwi localization is noted as being impacted by Klp10A knockdown. Can the authors speculate as to what Klp10A might be doing in relation to the other piRNA factors?

**Have all data underlying the figures and results presented in the manuscript been provided?**

Reviewer #1: Yes

Reviewer #2: Yes

PLOS authors have the option to publish the peer review history of their article (what does this mean?). If published, this will include your full peer review and any attached files.

Reviewer #1: No

Reviewer #2: No

---

## [Editor Report · Decision Letter 1]

3 Feb 2020

Dear Yukiko,

We are pleased to inform you that your manuscript entitled "A kinesin Klp10A mediates cell cycle-dependent shuttling of Piwi between nucleus and nuage." has been editorially accepted for publication in PLOS Genetics. Congratulations!

Yours sincerely,

Giovanni Bosco, Ph.D.

Associate Editor

PLOS Genetics

Gregory P. Copenhaver

Editor-in-Chief

PLOS Genetics

Comments from the reviewers (if applicable):

**Data Deposition**

http://datadryad.org/submit?journalID=pgenetics&manu=PGENETICS-D-19-01837R1

**Press Queries**

---

## [Editor Report · Acceptance letter]

4 Mar 2020

PGENETICS-D-19-01837R1 

A kinesin Klp10A mediates cell cycle-dependent shuttling of Piwi between nucleus and nuage. 

Dear Dr Yamashita, 

We are pleased to inform you that your manuscript entitled "A kinesin Klp10A mediates cell cycle-dependent shuttling of Piwi between nucleus and nuage." has been formally accepted for publication in PLOS Genetics! Your manuscript is now with our production department and you will be notified of the publication date in due course.

With kind regards,

Kaitlin Butler

PLOS Genetics

On behalf of:
